# Prevalence of Genetic Diamine Oxidase (DAO) Deficiency in Female Patients with Fibromyalgia in Spain

**DOI:** 10.3390/biomedicines11030660

**Published:** 2023-02-22

**Authors:** Gülşah Okutan, Eva Ruiz Casares, Teresa Perucho Alcalde, Guerthy Melissa Sánchez Niño, Bruno F. Penadés, Ana Terrén Lora, Lorena Torrente Estríngana, Sara López Oliva, Ismael San Mauro Martín

**Affiliations:** 1Research Centers in Nutrition and Health, CINUSA Group, 28036 Madrid, Spain; 2VIVO Laboratorio, Grupo Vivo, Alcobendas, 28100 Madrid, Spain; 3Department of Genetics, Faculty of Medicine, CEU-San Pablo University, Boadilla del Monte, 28668 Madrid, Spain; 4Faculty of Biological Sciences, Complutense University of Madrid, 28040 Madrid, Spain

**Keywords:** DAO enzyme, *AOC1*, fibromyalgia, variants, SNPs, genetic deficiency, histamine intolerance

## Abstract

Diamine oxidase (DAO) is an enzyme that metabolizes intestinal histamine. Single nucleotide polymorphisms (SNPs) of the Amine Oxidase Copper Containing 1 (*AOC1*) gene can lead to low enzymatic activity or functionality in histamine metabolism. This study aimed to determine the prevalence of DAO deficiency for four variants of the *AOC1* gene, p.Thr16Met (rs10156191), p.Ser332Phe (rs1049742), p.His664Asp (rs1049793), and c.691G > T (rs2052129), in 98 Spanish women with fibromyalgia between the ages of 33 and 60 years, and compare the distribution of allelic and genotypic frequencies with those of European population samples in Hardy–Weinberg equilibrium extracted from the Allele Frequency Aggregator (ALFA) database. The patients’ DNA was extracted, and analyzed using SNPE Multiplex (Single Nucleotide Primer Extension). The prevalence of genetic DAO deficiency was 74.5% based on the four variants of the *AOC1* gene. SNP deficits were found at frequencies of 53.1% for p.Thr16Met, 49% for c.691G > T, 48% for p.His664Asp, and 19.4% for p.Ser332Phe. The allele and genotypic frequencies of the women with fibromyalgia did not differ from the European population. Variants of the *AOC1* gene that are associated with genetic DAO deficiency could serve as a disruptive biomarker in patients with fibromyalgia. This study was registered in ClinicalTrials.gov Identifier: NCT05389761.

## 1. Introduction

Histamine is an organic nitrogenous compound that plays an important role in immune function, and the digestive and central nervous systems. An intolerance to histamine (HIT) might occur due to the release of histamine in the body or the inability to break down this compound. Various health conditions can contribute to a HIT. The underlying conditions for increased availability may include an endogenous histamine overproduction caused by allergies, mastocytosis, bacteria, gastrointestinal bleeding, or ingestion of foods with high contents of histamine or histidine. Biogenic amines, such as putrescine, may also play an important role in the cleavage of histamine from the mucosal mucin junction, resulting in an increase in free absorbable histamine. Nevertheless, the main cause of HIT is an impaired enzymatic histamine degradation caused by genetic or acquired impairment of the diamine oxidase (DAO) or histamine N-methyltransferase (HNMT) [1].

DAO deficiency may be related to certain pathologies, and histamine should be considered in the differential diagnosis of patients with various diseases [2], including those of un-known origin such as fibromyalgia syndrome. Fibromyalgia is a chronic pathology in which inflammation and pain symptoms may be observed due to the release of histamine molecules from mast cells [3]. It is a long-term condition characterized by widespread musculoskeletal pain accompanied insomnia, headache and anxiety, among many others symptoms. Despite recent advances, the lack of an accurate diagnosis and treatment remain barriers to the recognition of this disease [4].

Fibromyalgia treatment is usually multidisciplinary and includes a combination of drug therapies. The components in these drugs may also inhibit DAO enzymatic activity. [5]. Furthermore, dysfunction of the control of biogenic amines in the central nervous system can result in fibromyalgia with pain and inflammation. [6]. Indeed, the prevalence of reduced DAO levels in individuals with HIT is associated with headache, gastrointestinal disorders, and skin symptoms [7], and these symptoms are also present in fibromyalgia patients.

According to the International Society of DAO Deficiency, DAO deficiency can cause various adverse effects on various systems of the body, such as respiratory (nasal congestion, asthma); cardiovascular (hypotension, hypertension, arrhythmias); central nervous (migraine, headaches, hangover, dizziness); digestive (irritable bowel syndrome, constipation, satiety, stomach pain, vomiting); muscular (fibromyalgia, muscle pain); and skeletal (osteopathic pain) [8,9]. Fibromyalgia is a complex and common condition in the clinic causing chronic pain, and is a major cause of morbidity; the prevalence of FM in the general population was estimated at between 0.5 and 5% [10]. It can be accompanied by several other psychological symptoms, including sleep disorders, morning stiffness, headaches, and memory problems [11], and these findings suggest that histamine may also play a role in the clinical symptoms of fibromyalgia and trigger an increase in low-grade inflammation [12]. The gene encoding DAO (*AOC1*) is located on chromosome 7 (7q34-q36) in the human genome. A total of 85 SNPs are located in the human *AOC1* gene. Seven of these SNPs produce an amino acid substitution, serving as candidates that cause alterations in the metabolic capacity of the enzyme. Of the polymorphisms found in the *AOC1* gene sequence, four have been associated with a low enzymatic activity in the metabolism of histamine [13,14]. This relationship has been explained by symptoms due to ingested histamine; however, it has not been proven with a specific diagnosis. In this sense, population genetics is important to understand the genetic composition, biological mechanism, diagnosis, and the gene-diet interconnections and their effects in individuals. For instance, lactase persistent or lactase non-persistent is genetically determined, and primary lactose intolerance is associated with certain genotypes [15,16]. In other assessments, genome-wide association studies (GWAS) led to the successful identification of multiple genetic susceptibility variants to several complex human diseases using panels comprising hundreds of thousands of SNPs, and rapidly led to the discovery of strong and consistent associations with multiple complex diseases, including type 2 diabetes, stroke, obesity, and various types of cancer [17].

Based on the symptomatology and comorbidities associated with this condition, an increased frequency of acquired epigenetic and chromosomal alterations has been reported in women with fibromyalgia [18]. As a result, epigenetics may be convenient in identifying biomarkers that can be used for diagnosis and predicting clinical outcomes. These findings suggest that SNPs of the *AOC1* gene may be appropriate in the diagnosis of the clinical symptoms of fibromyalgia.

SNPs can be a reliable diagnostic method as the serum DAO level is inadequate for many factors such as sex, menstrual cycle, certain foods, and medications [19]. Owing to the high prevalence and remarkable clinical and social impact of fibromyalgia, combined with the complexity of its treatment, healthcare professionals must be educated and trained on the proper treatment of patients with fibromyalgia. Recently, attempts have been made to search for new molecular biomarkers that could be used to identify new treatments and improve the pathophysiology of these patients [20]. Consequently, the present study aimed to determine the prevalence of genetic DAO deficiency associated with any of the four variants in the *AOC1* gene, namely, p.Thr16Met (rs10156191), p.Ser332Phe (rs1049742), p.His664Asp (rs1049793), and c.691G > T (rs2052129), in Spanish women with fibromyalgia, and compare the obtained distribution of allelic and genotypic frequencies with those of European population samples in Hardy–Weinberg Equilibrium (HWE) extracted from the Allele Frequency Aggregator (ALFA) database [21].

## 2. Materials and Methods

### 2.1. Study Design

An epidemiology study was conducted to observe the association between genetic DAO deficiency in the Spanish female population with fibromyalgia. The study was conducted at the Research Center in Nutrition and Health (CINUSA Group, Madrid, Spain) in CINUSA’s Clinic and Ruber International Hospital (Paseo de la Habana 43, Madrid, Spain).

### 2.2. Sample

The sample population consisted of 98 Spanish women with fibromyalgia, between the ages of 30 and 60 years. Age and medical diagnosis were considered in the inclusion criteria, whereas multiple chemical sensitivity, a diet low in histamine, DAO treatment, and some chronic diseases (diabetes, cancer, AIDS, and Alzheimer’s) were considered as exclusion criteria.

European population samples were extracted from the ALFA database [21] for each SNP included in the study (p.Thr16Met: *n* = 269740; p.Ser332Phe: *n* = 275948; p.His664Asp: *n* = 96876; c.691G > T: *n* = 223202).

### 2.3. AOC1 Variant Genotyping

Four variants in the *AOC1* gene were analyzed for each sample: p.Thr16Met (rs10156191), p.Ser332Phe (rs1049742), p.His664Asp (rs1049793), and c.691G > T (rs2052129).

Oral mucosa sampling was performed by rubbing the inside of the cheek with a sterile cotton swab (Sarstedt), according to the standard hygiene protocol that consists of keeping the patient’s mouth clean for at least 60 min before taking the sample. Two samples of buccal mucosa were taken from each patient and DNA was isolated using an automatic isolation procedure in the QIASymphony SP platform (QIAGEN, Hilden, Germany) with QIASymphony DSP DNA Mini Kit (QIAGEN). Genotyping of the samples was performed using Multiplex SNPE (Single Nucleotide Primer Extension) followed by capillary electrophoresis in an ABI 3500 Genetic Analyzer (Thermofisher Scientific, Applied Biosystems, Waltham, MA, USA) at VIVOLABS. The methodological recommendations of the American College of Medical Genetics (ACMG, Bethesda, MD, USA) were followed to obtain the most accurate results [22].

### 2.4. Statistical Analysis

Descriptive analysis and frequency comparisons via Pearson’s Chi-square exact test were performed using IBM SPSS Statistics 25 (IBM Corporation, Armonk, NY, USA). Previously, the European population samples for each variant were confirmed to be in HWE using R software (v4.1.2). The contrasts were bilateral with a confidence level of 95%.

### 2.5. Regulatory Approval

The study was approved by the Research Ethics Committee with Medications (CEIm) Hospital Group Quirónsalud-Catalunya (Barcelona, Spain; approval number, nº24/2021). The study was also registered in ClinicalTrials.gov Identifier: NCT05389761.

This study was carried out in accordance with the latest versions of the Declaration of Helsinki (World Medical Association, 2013), the GCP standards (ICH 2016 R2), and the legal standards and regulations for biomedical research in humans (Law 14/ 2007 and RD 1090/2015).

## 3. Results

The sample population consisted of 98 women between 33 and 60 years old (48.5 years ± 7.5). The prevalence of genetic DAO deficiency was 74.5% after the analysis of the four variants of the *AOC1* gene. This value represented all patients with at least one minority or risk allele, that is, associated with DAO deficiency, in any of the variants analyzed. The genetic deficit associated with each of the analyzed SNPs had the following frequencies: p.Thr16Met (53.1%), c.691G > T (49%), p.His664Asp (48%), and p.Ser332Phe (19.4%).

The allelic and genotypic prevalence had similar distributions for the SNPs analyzed, except for p.Ser332Phe (Figure 1). Patients had a DAO-normal activity allele of approximately 70%; however, the activity of p.Ser332Phe exceeded 90%. Genotypic frequencies of approximately 50% were observed for the homozygous genotype with the DAO-normal activity allele and frequencies close to 10% were observed for the homozygous genotypes with the DAO-deficiency associated allele. In contrast, the TT genotype was not found for p.Ser332Phe, while the frequencies of the homozygous genotype exceeded 80%.

To verify whether the distribution of the allele and/or genotypic frequencies of the sample deviated from a population in HWE, samples from the European population were used by accessing the ALFA database. These samples satisfactorily met the HWE assumption for all SNPs tested (*p* = 1). Although it could be observed that the frequencies of the minority alleles, associated with DAO deficiency, were more frequent in the sample with fibromyalgia, it could not be confirmed that there were differences between the samples (Table 1).

Finally, 14 genotype combinations were observed among the variants included in the study for women with fibromyalgia (Figure 2). The combination of genotypes with the allele associated with the normal activity of the DAO enzyme (25.5%) was found to be predominant, followed by the reduced DAO activity-associated alleles p.Thr16Met and c.691G > T in the heterozygous state (15.3%), and the heterozygous reduced DAO activity-associated genotype, p.His664Asp (11.2%). The prevalence of the remaining combinations was found to be below 10% in our sample; however, the prevalence for many combinations can vary in the population within the expected margins.

## 4. Discussion

In this study, four variants of the *AOC1* gene were evaluated: p.Thr16Met (rs10156191), p.Ser332Phe (rs1049742), p.His664Asp (rs1049793), and c.691G > T (rs2052129), whose minor allele is associated with lower levels of DAO activity and therefore a lower capacity for histamine degradation [13,14]. In this sense, our results revealed that 74.5% of the women with fibromyalgia had a genetic DAO deficiency, that is, they had at least one risk allele in one of the four variants analyzed.

In addition, allele and genotypic frequencies were compared with HWE European population samples drawn from ALFA, and no significant differences were observed in any of the SNPs. This could be due to the fact that the variants analyzed individually do not have the same effect as the accumulation of risk alleles associated with DAO deficiency. That is, a combination of genotypes that include the minor allele associated with less enzymatic activity. Although it is true that there were no differences between the samples compared, it was observed that the sample of women with fibromyalgia had a higher frequency of the allele associated with DAO deficiency in all the variants, except p.His664Asp.

The *AOC1* gene variants associated with low enzyme activity may be associated with fibromyalgia symptoms, like many neurological or digestive disorders. In another study conducted in Spain with 197 patients with migraine and 245 healthy controls, *AOC1* SNP rs10156191, which is linked to decreased DAO enzyme activity, was found to be associated with the risk of developing migraine, especially in women [23]. In a more recent study, serum DAO activity was found to be increased in patients with migraine, whereas the serum histamine levels were normal, and the polymorphisms were not associated with the risk for migraine as serum DAO activity was assessed [24]. Histamine has also been implicated in the pathogenesis of migraine. A prior study investigating the possible association between functional single nucleotide polymorphisms in the DAO gene in the chromosome 7q36.1 with the risk for migraine reported a lack of association between Thr105Ile SNP (rs1801105) polymorphism in the HNMT gene located at chromosome 2q22.1 [23].

These findings can help diagnose the susceptibility of fibromyalgia patients to certain symptoms according to the genetic results, and improve these symptoms through genetic diagnosis using enzyme supplementation and appropriate dietary therapy. However, the medications used by fibromyalgia patients, additional supplements, or their special diet should be registered and considered to understand their effects on symptoms and the results of the study.

In addition to migraine, DAO deficiency is associated with many pathologies, such as schizophrenia. In single-locus analyses, the frequency of the C allele of a novel SNP rs55944529 located at intron 8 was found to be significantly higher in the original large patient sample. This allele was also associated with a higher level of DAO mRNA expression in the Epstein–Barr virus-transformed lymphocytes. As the haplotype structure of the DAO genomic region differs between Caucasian and Asian populations, the associated genomic region may be different in different ethnic samples, which is an important aspect considered in this study [25].

Another symptom in patients with fibromyalgia is digestive disorders. Digestive disorders are related to DAO because it is synthesized only in the epithelial cells of intestinal villi, and a high concentration in the blood is tightly linked to abnormal intestinal barrier function. Accordingly, digestive disorders can be used as sensitive and accurate markers for monitoring activity [8,26]. This relationship is a useful tool in diagnosing HIT; this is because it demonstrates the benefit of a histamine free diet on treatment of the disease. A recent study evaluated the improvement of digestive symptoms in fibromyalgia patients following a diet modification according to histamine release test. The study revealed that diet modification improves certain clinical parameters related to the symptoms of the digestive sphere compared to the control diet [27].

Finally, the present study was not without its limitations. Instead of having a control group without fibromyalgia, but with characteristics similar to the sample drawn from the Spanish population, we used samples from the European Population in HWE drawn from ALFA. Likewise, the fact of comparing the allelic and genotypic frequencies of four SNPs individually with the European population seriously limited the obtaining of substantial results, taking into account that the impact of each variant is very small. Nevertheless, the cumulative effect of the variants with the minor allele, which are associated with DAO deficiency, could offer very interesting results. Furthermore, a relatively small sample size was used, although, admittedly, the idea was to provide proof of concept, as DAO enzyme activity had not previously been estimated in the population of women with fibromyalgia. Without forgetting that serum DAO data were not collected, which could have reinforced the findings of the genetic test and the assumption of considering that they had low DAO activity when women had at least one risk allele.

For all these reasons, future research should be aimed at recruiting more subjects with fibromyalgia and analyzing the cumulative effect of the risk alleles for the selected variants, as well as using genetic and serum analysis tools to observe the enzymatic behavior according to the presence of associated alleles to DAO deficiency in the population with fibromyalgia.

## 5. Conclusions

In this study, we found a high prevalence of genetic DAO deficiency (that is, the presence of one or more alleles associated with low DAO activity in the four SNPs of the *AOC1* gene analyzed) in Spanish women with fibromyalgia. The allele and genotypic frequencies of this sample did not differ from the European population in HWE and more research is required to observe the cumulative effect of risk alleles and to study the potential offered by the *AOC1* gene as a biomarker of DAO enzymatic activity in the fibromyalgia population.

## Figures and Tables

**Figure 1 biomedicines-11-00660-f001:**
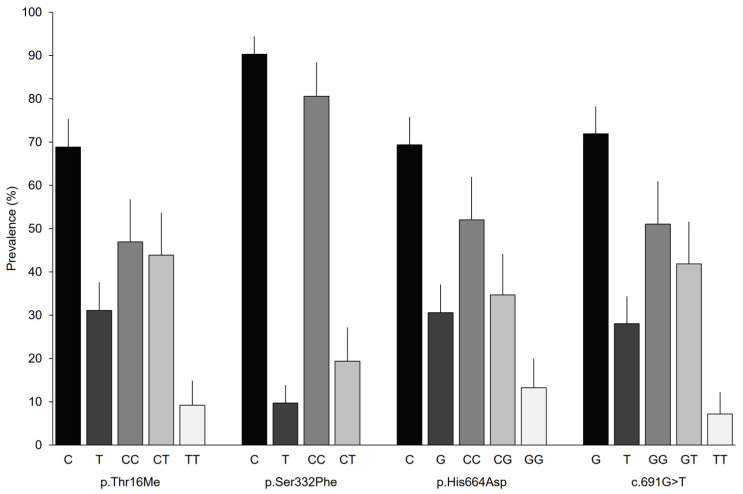
Allelic and genotypic prevalence based on the analyzed variant in the *AOC1* gene. Error bars represent 95% CIs.

**Figure 2 biomedicines-11-00660-f002:**
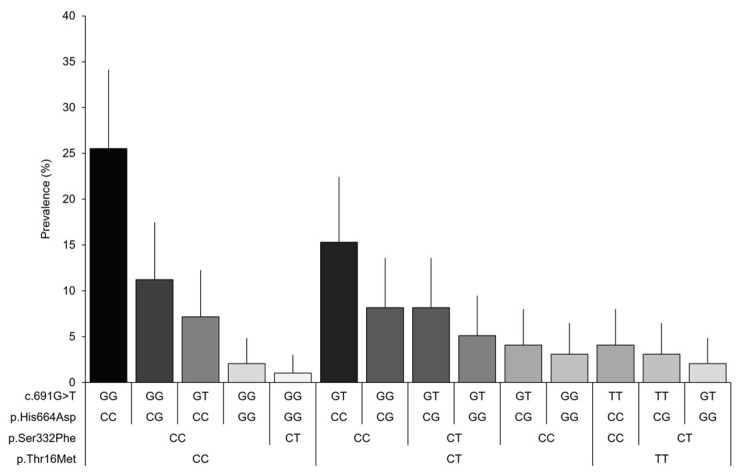
Genotypic prevalence based on the combination of the analyzed variants of the *AOC1* gene. Error bars represent 95% CIs.

**Table 1 biomedicines-11-00660-t001:** Allelic and genotypic frequencies by variant for patients with fibromyalgia and the European population in Hardy–Weinberg equilibrium.

	Alleles *n* (%)			Genotypes *n* (%)			
p.Thr16Met (rs10156191)	C	T	*p*	CC	CT	TT	*p*
FB (*n* = 98)	135 (68.9)	61 (31.1)	0.122	46 (46.9)	43 (43.9)	9 (9.2)	0.273
EP-HWE (*n* = 269740)	199,429 (73.9)	70,311 (26.1)		147,446 (54.7)	103,967 (38.5)	18,327 (6.8)	
p.Ser332Phe (rs1049742)	C	T		CC	CT	TT	
FB (*n* = 98)	177 (90.3)	19 (9.7)	0.212	79 (80.6)	19 (19.4)	0 (0)	0.186
EP-HWE (*n* = 275948)	255,950 (92.8)	19,998 (7.2)		237,401 (86)	37,098 (13.4)	1449 (0.5)	
p.His664Asp (rs1049793)	C	G		CC	CG	GG	
FB (*n* = 98)	136 (69.4)	60 (30.6)	0.938	51 (52)	34 (34.7)	13 (13.3)	0.181
EP-HWE (*n* = 96876)	67,537 (69.7)	29,339 (30.3)		47,084 (48.6)	40,907 (42.2)	8885 (9.2)	
c.691G > T (rs2052129)	G	T		GG	GT	TT	
FB (*n* = 98)	141 (71.9)	55 (28.1)	0.153	50 (51)	41 (41.8)	7 (7.1)	0.334
EP-HWE (*n* = 223202)	170,473 (76.4)	52,729 (23.6)		130,200 (58.3)	80,545 (36.1)	12,457 (5.6)	

Note: FB, Fibromyalgia P NoFFB, patients with Fibromyalgia; and EP-HWE, European Population in Hardy–Weinberg Equilibrium.

## Data Availability

The data presented in this study are available on request from the corresponding author.

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
