# Peer review of "Prevalence of Genetic Diamine Oxidase (DAO) Deficiency in Female Patients with Fibromyalgia in Spain"

_biomedicines, 2023, doi:10.3390/biomedicines11030660_

Round 1

Reviewer 1 Report

The idea of the study is correct and seems to be justified in regard to the current state of the art. I don't have any substantial concerns as well from technical point of view the are no significant flaws. 

My remarks are rather focused on editorial aspects which are weak elements and their fixation in my opinion would improve significantly revised paper.

1. Figures are in my opinion used badly. Figure 1 is useless. 75 vs. 25 % maybe as well just mentioned in the text and it would be sufficient...

2. Real prevalence should be presented on cake diagrams as figure 2 and 3, those percentages are not that visible on the bars…

3. In the methods paragraph there are missing data on particular reagents used in the study. It is standard that it should be presented in the reproducible manner…

4. References should be refreshed and revised deeply. There are old papers, what in the perspective of described prevalence shouldn't be present. Moreover position 8 and 12 are the same. I would change position 5 into origin reference for presented thesis.

5. "Funding: This work was supported by DR Healthcare-AB Bioteck. The authors declare that no funds, grants, or other forms of support were received during the preparation of this manuscript." This is not clear for me. DR Healthcare bought reagents, or will pay for laboratory tests etc. Because it sounds that supported but didn't pay for anything…

Reviewer 2 Report

This study investigated the prevalence of dao deficiency in female patients with fibromyalgia in Spain. The rational behind the article was clear and straight forward. The manuscript is almost well written. 

The authors should mentioned in the method section of the abstract more details about how patients were chosen.

While many different sources are used to set up the study in the introduction, little previous evidence is stated. The introduction is thus short and poorly sets up the rationale for the study. More attention to how this study fits into previous work in fibromyalgia and inflammation should be added to improve this section. Please refer to doi:

10.3390/ijms22126471

There are some minor grammar issues that should be fixed in order to aid the accessibility of the results to

the reader.

Reviewer 3 Report

In the current study, the authors have examined the prevalence of genetic diamine oxidase deficiency in female fibromyalgia patients. Considering the complexity of fibromyalgia as a disease, any work that can advance the understanding of this disease is important.

While the manuscript was well-written overall, with some interesting findings, there are a few issues:

1) the introduction does not clearly explain the rationale behind the study of diamine oxidase gene in fibromyalgia

2) considering DAO deficiency could be a factor, it would be appropriate to present data for serum DAO levels in the subjects and those of controls to confirm genotype-phenotype correlation

3) While the authors have claimed that there is a high prevalence of genetic DAO deficiency in the current cohort, it is of concern that the differences are not statistically significant (Table 1). 

Round 2

Reviewer 3 Report

The reviewer would like to thank the authors for their responses, it is much appreciated. Just a few comments:

1) While in the manuscript it was stated that the aim of the study is to determine the prevalence of DAO deficiency for four variants of the AOC1 gene (Line 16, 167), the authors have pointed out that DAO levels can be affected by factors such as gender, menstrual cycle and food and medications. This rather detract from the whole aim of the study

2) Since the authors have stated that "it was expected that the allele and genotypic frequencies of the SNP in our samples would not differ from the equilibrium of the European population because the effect of each SNP is small and the interest would be in the combination of several SNPs with risk alleles", wouldn't  it make more sense if the data was presented in such as way to reflect this?

Author Response

Dear Reviewer,

Please see the attachment for the answers to your comments and suggestions.

Apart from that, we would like to inform you that we have corrected some minor errors in the manuscript. These mistakes are related to references order, that is to say, we have changed the number of some references in order to put them in the correct order. Finally, we have added the correct information about “Data Availability statement” as “The data presented in this study are available on request from the corresponding autor”.

If you need the version with change control, please do not hesitate to tell us and we will send you by email

We look forward to hearing from you regarding our submission and responding to any further questions and comments you may have.

Yours sincerely

Round 3

Reviewer 3 Report

The author would like to thank the authors for clearing up the concerns associated with the manuscript. Despite the limitations of the study, the work here will provide groundwork for further work.